https://doi.org/10.1038/s41467-019-12946-4　　**OPEN**

# Rice production threatened by coupled stresses of climate and soil arsenic

E. Marie Muehe[1,3]*, Tianmei Wang[1], Carolin F. Kerl [2], Britta Planer-Friedrich[2] & Scott Fendorf [1]*

Projections of global rice yields account for climate change. They do not, however, consider the coupled stresses of impending climate change and arsenic in paddy soils. Here, we show in a greenhouse study that future conditions cause a greater proportion of pore-water arsenite, the more toxic form of arsenic, in the rhizosphere of Californian *Oryza sativa* L. variety M206, grown on Californian paddy soil. As a result, grain yields decrease by 39% compared to yields at today's arsenic soil concentrations. In addition, future climatic conditions cause a nearly twofold increase of grain inorganic arsenic concentrations. Our findings indicate that climate-induced changes in soil arsenic behaviour and plant response will lead to currently unforeseen losses in rice grain productivity and quality. Pursuing rice varieties and crop management practices that alleviate the coupled stresses of soil arsenic and change in climatic factors are needed to overcome the currently impending food crisis.

[1] Earth System Science Department, Stanford University, Stanford, CA 94305, USA. [2] Environmental Geochemistry, Bayreuth Center for Ecology and Environmental Research (BayCEER), University of Bayreuth, D-95440 Bayreuth, Germany. [3] Present address: Center for Applied Geoscience, University of Tuebingen, Tuebingen, Germany. *email: eva-marie.muehe@uni-tuebingen.de; fendorf@stanford.edu

With more than half of the world's population depending on rice for subsistence[1], it is crucial to ensure future rice production. Current empirically derived and modelled projections on rice production account for climate change as a constraining factor to yields[2–7]. However, they do not consider the coupled stresses of climate change and the presence of arsenic, a plant and human toxin, in paddy soils[6–11].

The two determinants of climate change, elevated atmospheric $CO_2$ and temperature, have opposing effects on plant growth and performance. Increasing atmospheric $CO_2$ stimulates the photosynthetic rate of rice, increasing water and nutrient use efficiency and allocation of more photosynthetic product to the root, ultimately resulting in greater biomass and increased yields[2,12–15]. Increasing temperature above the plant's optimum, contrastingly, decreases photosynthetic rate, thereby reducing biomass yields[2,12–14]. Temperature often overpowers positive $CO_2$ effects, decreasing grain yields as a result of diminished grain filling[2,12–14]. Besides affecting crop yields, climatic parameters also affect nutrient and contaminant behaviour in soils[16–20], thereby potentially altering the extent of their uptake into crops. The most relevant contaminant in flooded rice paddies is arsenic—it occurs ubiquitously, and naturally, in paddy soils globally, and is often found in groundwater used for irrigation[21–23]. Furthermore, paddy soils irrigated with arsenic contaminated groundwater accumulate arsenic with each successive irrigation cycle, exacerbating the effects of contamination with time[10,22,24–26]. Once in the soil, flooding induces reductive dissolution of iron oxyhydroxide minerals[27,28] and reduction of arsenic adsorbed on soil minerals[29], increasing dissolved concentrations, and thus, plant availability of arsenic. Furthermore, sulphate-reducing and other methylating microbial communities in the soil methylate arsenic in a stepwise and reductive manner to mainly monomethylarsonic acid (MMAs(V)), dimethylarsinic acid (DMAs(V)), and trimethylarsine oxide (TMAs(V)O), which accumulate in rice grain[30–34]. Methylated arsenic species are taken up more slowly into rice compared with inorganic species, but they are more readily translocated to the grain[32–36]. Once taken up, arsenic diminishes plant growth and development, reducing grain yields and inhibiting both panicle development and grain filling[9–11,37]. Even though methylated arsenic is carcinogenic, it is considered a less toxic variant of arsenic[38], and thus international regulation of arsenic contents in foodstuff only consider inorganic arsenic.

Increased temperature, and to a lesser extent increased atmospheric $CO_2$, will affect soil biogeochemical processes by altering microbial community dynamics and activity and geochemical reactions that include contaminant/nutrient adsorption/desorption and mineral dissolution/precipitation[18,39–41]. How such altered biogeochemical processes affect soil arsenic and the resulting impacts on rice production, however, remain unresolved. Using rhizoboxes, Neumann et al.[16] showed that increased soil temperature affects rhizosphere dynamics, but grain production and quality were not able to be easily evaluated. The coupled stresses of a changing climate and soil arsenic on future rice yields and grain quality thus remain a serious gap in our knowledge of threats to food security. Here, we examine the combined, and potentially coupled, impacts of soil arsenic and climate change on rice grain yields and grain toxin levels. Our findings serve as a sentinel for the potential threats to future rice production.

For this study, soils from the rice-growing region of California, USA, and the associated rice variety, *Oryza sativa* L. medium-grain Calrose cultivar M206 was utilised to examine the potential compounding impacts of soil arsenic and a changing climate on rice production. Three growth factors were varied in a multifactorial design—soil arsenic concentrations, temperature, and atmospheric $CO_2$ concentrations. Soil arsenic concentrations varied from a background value of 7.3 mg As kg$^{-1}$ dry soil to 24.5 mg kg$^{-1}$ (see Methods and Supplementary Table 1). The higher concentration resides at the lower end of reported elevated values for Californian and Asian paddy soils[24,42] and represents a concentration change expected from irrigating with arsenic-bearing groundwater[10,22,24,25]. For climate variables, we examined temperatures of 33 and 38 °C and atmospheric $CO_2$ concentrations of 415 and 850 ppmv, where 33 °C and 415 ppmv $CO_2$ represent climatic conditions commonly found in Californian (U. S. climate data) and many Asian lowland rice-growing regions (Bangladesh Climate Data Portal). According to the International Panel on Climate Change (IPCC) report (5th Assessment Report, Representative Concentration Pathway RCP 8.5)[43], a 2014 projection indicated that it is very likely that atmospheric $CO_2$ concentrations would reach 570 ppmv compared with the present level and that temperature would increase 1.5 to 2 °C by the end of this century. However, a 2017 update changed the most likely outcome to the worst-case climate projection of 2014[44], which estimates a 5 °C higher temperature and doubled atmospheric $CO_2$. We therefore investigated what is considered the most likely climatic conditions for the end of the century (5 °C increase in temperature and 850 ppmv $CO_2$) on rice production. We used a greenhouse in greenhouse pot study design (see the Methods section, Supplementary Fig. 1) allowing for a tight control of temperature and atmospheric $CO_2$ concentrations, and to more easily determine differences in soil arsenic dynamics and plant behaviour.

While providing tight climatic control, our experimental approach imposes a restricted rooting depth and volume of water exchange (see discussion on utilising a greenhouse pot approach over other options in Supplementary Discussion 1). As a consequence, biogeochemical processes increasing (or decreasing) pore-water concentrations will be amplified with concomitant impacts on rice plants. Utilising soil arsenic concentrations within the lower range of documented levels for major rice-growing regions helps mitigate amplified arsenic concentrations resulting from reductive dissolution. Further, we used the same soil with two different arsenic contents to minimise biases towards differences in soil properties. Previous work demonstrated increased straw arsenic levels in pot studies, but similar grain arsenic contents compared with field trials[24,37,42,45–49]. Our study design followed Dittmar et al.[45], who had compared pot and field studies with the same soil and found similar arsenic contents in straw and grain of rice. Moreover, we compare our rice yields for varying climatic conditions to field trials to ensure comparable outcomes. Nevertheless, our findings may not represent quantitative trends for field production; they will, however, provide critical information on the processes impacting future rice production globally and qualitative (direction of) changes to future yields and grain quality.

## Results

**Determinants of grain yield.** Under today's climatic conditions with a low soil arsenic content, the rice variety M206 produced $11.5 \pm 1.2$ g of grain plant$^{-1}$ (Fig. 1) with grain filling of ~90% (Supplementary Fig. 2a). A shift to future climatic conditions (38 °C, 850 ppmv $CO_2$) alone resulted in a 16% yield loss ($9.6 \pm 1.2$ g of grain plant$^{-1}$) with 76% grain filling, while increased total soil arsenic alone caused a yield loss of almost 40% ($6.9 \pm 0.9$ g of grain plant$^{-1}$) with 96% grain filling. The combined impacts of changing climatic conditions and increased soil arsenic resulted in a 42% decrease in yield to $6.6 \pm 0.5$ g of grain plant$^{-1}$ with 81% grain filling. The interaction between soil arsenic and climatic conditions gave a *p*-value of 0.07 with ANOVA analysis, indicating a non-significant or weak interaction between soil arsenic

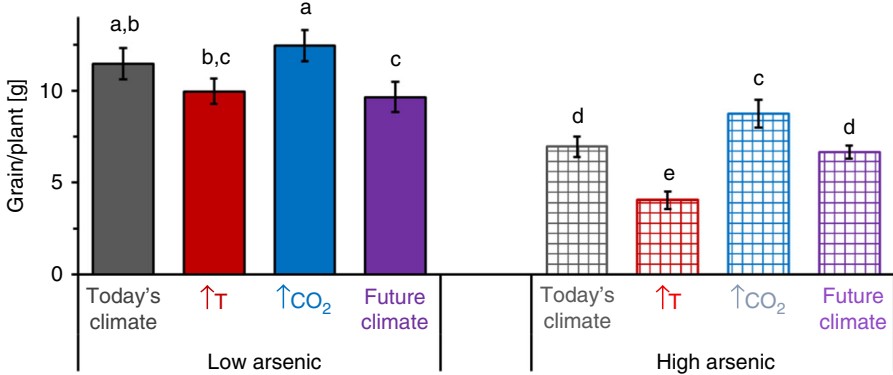

**Fig. 1** Grain yield of *Oryza sativa* L. cv M206 grown under different climatic and soil arsenic conditions. Dehusked grain weight per plant for rice grown under today's (grey, 33 °C and 415 ppmv $CO_2$), future (purple, 38 °C and 850 ppmv $CO_2$), elevated $CO_2$ (blue, 33 °C and 850 ppmv $CO_2$), and elevated temperature (red, 38 °C and 415 ppmv $CO_2$) climatic conditions on paddy soil with low (solid, 7.3 mg As kg$^{-1}$ dry soil) and high arsenic levels (chequered, 24.5 mg As kg$^{-1}$ dry soil). (Eight biological replicates, mean values ± standard errors were compared with each other using the unpaired *t* test at a 95% confidence interval. Different lowercase letters indicate that mean values were significantly different from each other ($p < 0.05$), for details see Supplementary Table 2a and Supplementary Fig. 2)

and climatic condition impacts on grain yield. The two climate factors, elevated temperature and elevated atmospheric $CO_2$, have opposing impacts on grain yield. Elevated temperature decreased grain yield by 13% (10.0 ± 0.7 g of grain plant$^{-1}$) with 83% grain filling, while elevated atmospheric $CO_2$ increased grain yield by 9% (12.4 ± 0.9 g of grain plant$^{-1}$) with 85% grain filling. In combination with arsenic, elevated temperature decreased grain yield by 65% (4.0 ± 0.7 g of grain plant$^{-1}$) with 58% grain filling, while elevated $CO_2$ resulted in a loss of 24% (8.7 ± 1.1 g of grain plant$^{-1}$) with 91% grain filling.

Under today's climatic conditions with a low soil arsenic content, variety M206 had 5.9 ± 0.2 panicles per plant (Supplementary Fig. 2b). A shift to future climatic conditions alone did not affect the number of panicles per plant, while total soil arsenic alone decreased them. The combined impacts of changing climatic conditions and increased soil arsenic decreased the panicle number to the same value as soil arsenic. The two climate factors (elevated temperature and elevated atmospheric $CO_2$) had opposing impacts on the number of panicles per plant. Under low and high soil arsenic conditions, elevated temperature decreased, while elevated atmospheric $CO_2$ increased, the number of panicles per plant. Under today's climatic conditions with a low soil arsenic content, 102 ± 6.3 spikelets were produced per panicle, which was adversely affected by an increase in soil arsenic (Supplementary Fig. 2c).

Unfilled spikelets of variety M206 contained 0.70 ± 0.07 µg As kg$^{-1}$ spikelet when grown under today's climatic conditions with a low soil arsenic content (Supplementary Fig. 2d). A shift to future climatic conditions alone increased the arsenic content of unfilled spikelets, while total soil arsenic alone increased the arsenic content of unfilled spikelets even more. The combined impacts of changing climatic conditions and increased soil arsenic increased the amount of arsenic in unfilled spikelets most. The two climate factors elevated temperature and elevated atmospheric $CO_2$ impacted spikelet filling oppositely. Under low and high soil arsenic conditions, elevated temperature increased, while elevated atmospheric $CO_2$ did not affect the arsenic content of unfilled spikelets. Under today's climatic conditions with a low soil arsenic content, the individual grain weight of M206 was 23.56 ± 0.36 mg (Supplementary Fig. 3). A shift to future climatic conditions alone did not affect individual grain weight, same as an increase in total soil arsenic alone. The combined impacts of changing climatic conditions and increased soil arsenic did also not affect individual grain weight. The two climate factors,

elevated temperature and elevated atmospheric $CO_2$, impacted individual grain weight oppositely. Under low and high soil arsenic conditions, elevated temperature decreased, while elevated atmospheric $CO_2$ slightly increased individual grain weight.

The harvest index indicates how much energy is invested in vegetative compared with reproductive growth. Under today's climatic conditions with a low soil arsenic content, the harvest index of variety M206 was 0.5 ± 0.0 grain to green biomass (Supplementary Fig. 4). A shift to future climatic conditions alone decreased the harvest index with an increase in straw biomass and decrease grain yield, while total soil arsenic alone did not affect the harvest index. The combined impacts of changing climatic conditions and increased soil arsenic decreased the harvest index to the same level as climatic change alone. The two climate factors elevated temperature and elevated atmospheric $CO_2$ again have opposite impacts on the harvest index. Under low and high soil arsenic conditions, elevated temperature decreased, while elevated atmospheric $CO_2$ hardly affected the harvest index of variety M206.

**Determinants of grain quality**. Under today's climatic conditions with a low soil arsenic content, dehusked but bran-containing grains from rice variety M206 contained a total of 393 ± 16.9 µg As kg$^{-1}$ grain, of which ~250 µg As kg$^{-1}$ grain was inorganic arsenite (Fig. 2; Supplementary Table 2). A shift to future climatic conditions alone increased the total grain arsenic content to 580 ± 21.0 µg As kg$^{-1}$ grain with the contribution of inorganic arsenic doubling in the grain to ~450 µg As kg$^{-1}$. With increased total soil arsenic alone, the total amount of arsenic in the grain increased to 821 ± 30.4 µg As kg$^{-1}$ grain with the contribution of inorganic arsenic remaining at ~250 µg As kg$^{-1}$ grain. The combined impacts of changing climatic conditions and increased soil arsenic resulted in a total grain arsenic increase to 1004 ± 17.9 µg As kg$^{-1}$ grain with ~400 µg As kg$^{-1}$ being inorganic arsenic. The interaction between soil arsenic and climatic conditions gave a significant *p*-value of 0.008 with ANOVA analysis, indicating that soil arsenic and climatic condition also impact grain quality interactively. The two climate factors, elevated temperature and elevated atmospheric $CO_2$, impacted grain arsenic content differently. Elevated temperature alone increased inorganic and organic grain arsenic content similarly to future climatic conditions, while elevated atmospheric $CO_2$ alone resulted in a similar inorganic and organic grain arsenic content

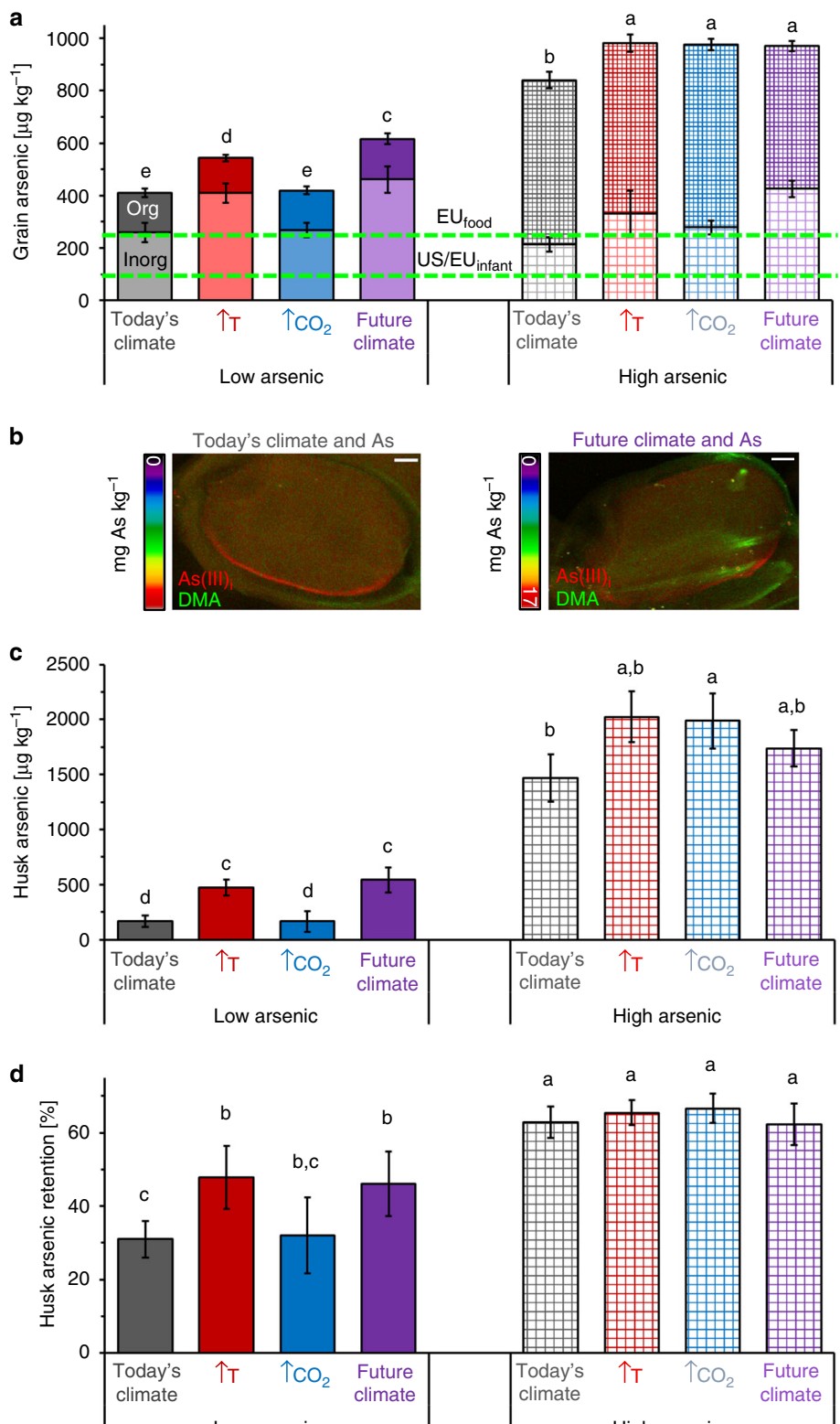

**Fig. 2** Arsenic contents in *Oryza sativa* L. cv. M206 grains produced under different climatic and soil arsenic conditions. **a** Amount of organic (upper bar) and inorganic (lower bar) arsenic accumulated in dehusked grains produced under today's (grey), future (purple), elevated $CO_2$ (blue) and elevated temperature (red) climatic conditions with low (solid) and high (chequered) soil arsenic levels. Upper green dashed line represents the European Union limit of 250 µg inorganic arsenic kg$^{-1}$ in husked rice, and the lower green dashed line represents the European Union and United States limit of 100 µg inorganic arsenic kg$^{-1}$ in infant food. **b** Arsenic species (arsenite and DMA) X-ray fluorescence micrographs of husked grains produced under today's and future climatic conditions with high soil arsenic levels. Scale bar 500 µm. **c** Amount of arsenic accumulated in husk, and **d** percentage of arsenic retained by husk compared to amount of arsenic that entered the grain. (Eight biological replicates, mean values ± standard errors were compared to each other using the unpaired *t* test at a 95% confidence interval. Different lowercase letters indicate that mean values were significantly different from each other (*p* < 0.05), for details see Supplementary Table 2b, c, d)

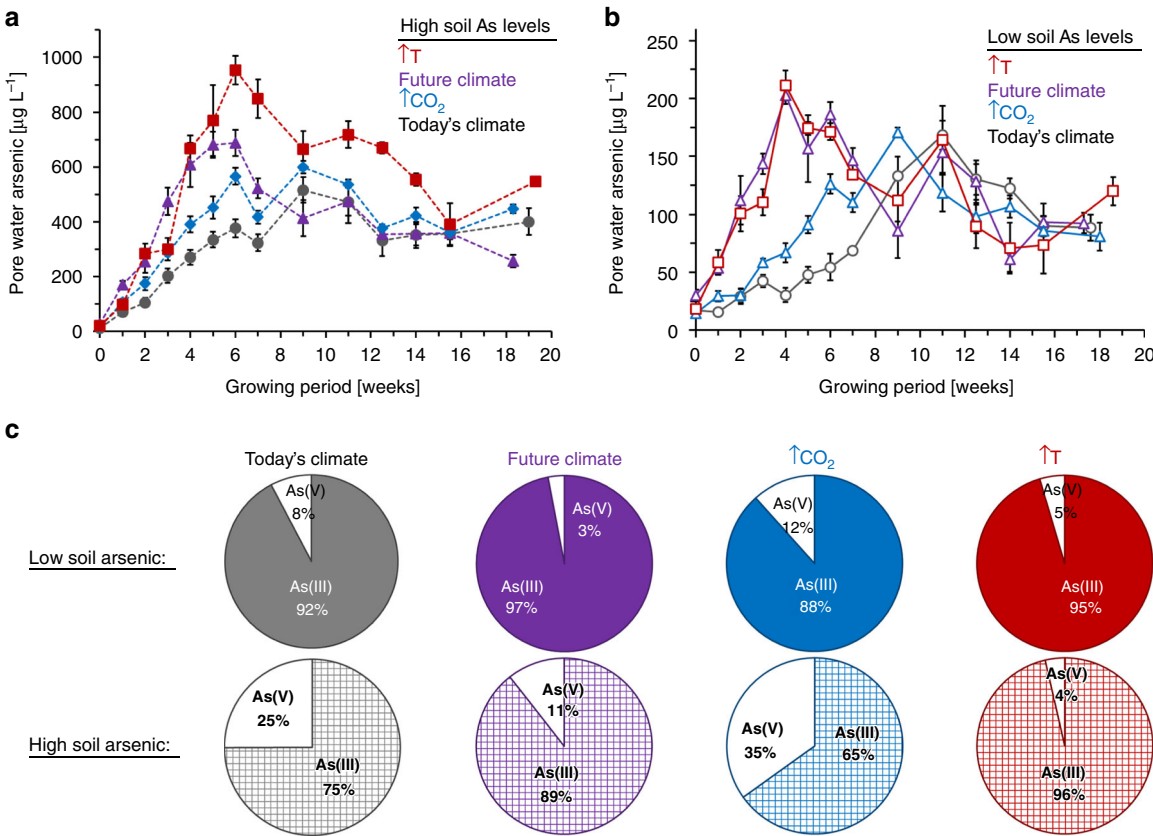

**Fig. 3** Dissolved arsenic in rhizosphere pore-water of *Oryza sativa* L. cv. M206 under different climatic and soil arsenic conditions. The total pore-water arsenic under (**a**) high soil arsenic levels (filled symbols with dashed line) and (**b**) low soil arsenic levels (empty symbols with solid line), **c** pore-water arsenite (here labelled as As(III)) and arsenate (here labelled as As(V)) contributions during grain filling. Pore-water was extracted at 10-cm soil depth from M206 grown under today's (grey), future (purple), elevated $CO_2$ (blue) and elevated temperature (red) climatic conditions. (Six biological replicates, mean values ± standard errors, see cumulative and rate values in Supplementary Table 4)

as grains produced under today's climatic conditions. In combination with arsenic, elevated temperature increased the inorganic grain arsenic fraction similarly as under future conditions, while elevated $CO_2$ showed a similar inorganic arsenic content as under today's climatic conditions.

Under today's climatic conditions with a low soil arsenic content, the husk contained $166 \pm 70.3$ µg As kg$^{-1}$, which made up 31% of the arsenic moving into the grain (Fig. 2). Either a shift to future climatic conditions alone or an increase in total soil arsenic content increased the arsenic content of the husk (alongside increased arsenic retention by the husk). The combined impacts of changing climatic conditions and increased soil arsenic enhanced the arsenic content of the husk to the same level as an increase in soil arsenic alone. The two climate factors, elevated temperature and elevated atmospheric $CO_2$, impacted the harvest index differently. Under low soil arsenic conditions, elevated temperature increased the arsenic content of the husk, while elevated atmospheric $CO_2$ did not affect the amount of arsenic in the husk. Under high soil arsenic conditions combined with elevated temperature or $CO_2$, no difference in arsenic content of the husk was observed compared with today's climate conditions. X-ray fluorescence imaging of the grains differentiate the arsenic species in the husk, bran and endosperm (Fig. 2; Supplementary Table 3). Under projected future climatic conditions, almost two times more dimethylarsinic acid (DMA) was retained by the husk compared with today's climatic conditions. Arsenite ratios in the bran to endosperm also increased under future climatic conditions in low and high soil arsenic settings.

**Rhizosphere arsenic and iron dynamics**. Three and a half times-higher total arsenic concentrations in the soil resulted in four-times higher dissolved pore-water concentrations of $4.12 \pm 0.89$ mg L$^{-1}$ compared with $1.04 \pm 0.12$ mg L$^{-1}$ (Fig. 3; Supplementary Table 4). Climatic change alone increased the dissolved pore-water concentration to just $1.65 \pm 0.09$ mg L$^{-1}$, although higher pore-water concentrations (and thus plant root exposure) occurred before week 8 in the growth cycle of the plant. Increasing temperature, and to a slight extent increasing atmospheric $CO_2$ concentrations, further exacerbated the partitioning of arsenic from solids to pore-water with respective $6.98 \pm 0.59$ and $5.13 \pm 0.40$ mg L$^{-1}$. Overall, temperature lead to highest dissolved arsenic exposure at earlier growth phases of the plant compared with elevated atmospheric $CO_2$ concentrations. After 8 weeks of plant growth, dissolved arsenic concentrations were similar for all climatic conditions, varying only with the total arsenic content in the soil.

In addition to an increase in dissolved arsenic concentrations and a shift in arsenic exposure timing under future climatic conditions, the proportion of arsenite, a trivalent species of arsenic more toxic to humans compared with the pentavalent form arsenate[50], in pore-water also changed (Fig. 3). While $92.3 \pm 8.7\%$ of pore-water arsenic was arsenite under today's low arsenic conditions, arsenite accounted for $97.0 \pm 6.0\%$ under future climatic conditions with low arsenic levels in the soil. At higher soil arsenic concentrations, $74.9 \pm 4.4\%$ of the arsenic was present as the arsenite species under today's climatic conditions (as expected as the soil was spiked with 70% arsenite and 30% arsenate), compared with $89.4 \pm 14.0\%$ under future climatic conditions. Temperature as the sole climatic variable shifted the

proportion of arsenite to 96.4 ± 6.2% of total pore-water arsenic, while elevated atmospheric $CO_2$ alone decreased the proportion of arsenite to 65.1 ± 3.5%.

Iron concentrations in the pore-water generally increased before week 8 in the growth cycle of rice M206, while it reached the same level under all climatic and soil arsenic conditions during the reproductive phase of the plant (Supplementary Fig. 5). Under low soil arsenic conditions, elevated temperature alone and combined future climatic conditions increased pore-water iron concentrations, with the timing of highest pore-water iron concentrations pushed towards the beginning of plant growth cycle. Under elevated atmospheric $CO_2$ conditions, the amount of iron in the pore-water also increased compared with today's climate, although the timing of iron release remained the same. At high soil arsenic conditions, the response of pore-water iron dynamics was less pronounced.

**Microbial and carbon dynamics**. We used 16S rRNA gene copy numbers $g^{-1}$ dry soil to approximate changes in microbial community cell numbers (Supplementary Fig. 6a). In comparison with today's climatic conditions with a low soil arsenic content, future climatic conditions with a low soil arsenic content resulted in an intermediate decrease in cell numbers. High soil arsenic levels alone decreased cell numbers compared with low soil arsenic conditions. Future climatic conditions combined with high soil arsenic levels exhibited similar cell numbers as under future climatic conditions alone. While increased temperature alone resulted in a strong decrease in cell numbers, elevated atmospheric $CO_2$ alone caused a slight decrease.

Dissolved organic carbon within the rhizosphere pore-water was impacted most appreciably by temperature (Supplementary Fig. 6b). Plants grown at elevated temperature or under future climatic condition had nearly twice as much DOC compared with the rhizosphere for today's climate or under elevated $CO_2$ conditions. High soil arsenic supressed the differences, where slightly greater DOC occurred for lower temperature conditions and slightly less DOC occurred at higher temperature relative to low soil arsenic conditions. Similarly, for low soil arsenic, dissolved inorganic carbon (DIC) was impacted most appreciably by temperature, being highest for elevated temperature conditions and then future climate conditions (Supplementary Fig. 6c). High soil arsenic altered the controlling influence of growth conditions; temperature effects alone were muted, but the combination of elevated $CO_2$ and increased temperature of future climate conditions resulted in the greatest DIC.

## Discussion

Under today's climatic conditions with a low soil arsenic content, the observed yield of rice variety M206 grown under greenhouse conditions is in accordance with reported performance in California[51]. A decrease in yield of ~16% under future climatic growth conditions (38 °C, 850 ppmv $CO_2$) alone has also been reported in empirically determined and modelled rice production studies[2–6], although these studies are not specific to rice variety M206. As previously reported, most of that loss is due to temperature stress, which can only partially be counteracted by elevated atmospheric $CO_2$. A loss of 39% grain yield due to increased total soil arsenic alone is also consistent with previous findings[37]. Critically, soil arsenic accounts for nearly twice as much grain loss as combined temperature and $CO_2$ changes, indicating that under conditions tested here, arsenic is a stronger determinant of yield than climate. Thus, the combined impacts of changing climatic conditions and increased soil arsenic cause a loss of 42% decrease in yield, which is not compensated by elevated $CO_2$.

Variations in grain yield under different growing conditions are driven by differing impacts of arsenic and climatic stresses on plant physiological traits and rhizosphere dynamics. Overall, soil arsenic (at the concentrations examined) did not affect individual grain weight; it did, however, decrease the number of panicles per plant and spikelets per panicle. By contrast, temperature decreased individual grain weight as observed previously[52], and reduced spikelet filling, especially when combined with soil arsenic stress. Elevated temperature combined with higher soil arsenic levels increased arsenic concentrations in unfilled spikelets, indicating that these arsenic levels inhibited grain filling in spikelets. Under climatic stress, variety M206 invested more energy in vegetative compared with reproductive growth, indicated by a lower harvest index, which was unaffected by soil arsenic alone. Soil arsenic had a principal effect of decreasing overall plant productivity.

The observed effects on rice performance with increased soil arsenic concentrations is caused by an increase in dissolved pore-water concentrations in the rhizosphere that lead to increased plant arsenic accumulation. Increasing temperature[14] further exacerbates the partitioning of arsenic from solids to pore-water, stimulating the reductive dissolution of As-bearing Fe(III) (hydr)oxides[17] with the concomitant increase in pore-water arsenite and iron concentrations. Thus, rice plants are exposed to higher arsenite concentrations at earlier growth stages. Increasing atmospheric $CO_2$ concentrations also stimulated microbial respiration in the rhizosphere (increased inorganic pore-water carbon) causing reductive dissolution of Fe(III) (hydr)oxides and arsenite oxidation[53], as indicated by increases in iron and arsenate concentrations.

Our study imposed a rapid temperature and $CO_2$ transition that would otherwise transpire over the coming decades. As a result, the microbial community may not evolve comparably despite most organisms having a rapid doubling time and prior acclimation of the soil to high arsenic and climate for a season. In our study, bacterial numbers decreased significantly due to a rise in temperature, which was only partially recovered by elevated atmospheric $CO_2$ under future climatic conditions. Nevertheless, the functional indicators of microbial activity (inorganic carbon pore-water concentrations of redox active elements) for arsenic are entirely consistent with expectations; an increase in temperature and/or atmospheric $CO_2$ will lead to increases in organic carbon exudation from plants and microbes (shown by increased organic carbon in pore-water) and increases in microbial activity[18,39,54] as noted by the DIC levels, and thus, greater oxygen demand and greater As(V)/Fe(III) reduction with increases in temperature[17].

Climate and arsenic imprints on grain yield are, however, only half of the story; their coupled impacts on grain quality (inclusive of toxin and nutrient content) also are critical for consideration. Dehusked but bran-containing grains from rice grown under current climatic and soil arsenic conditions contained a total of 393 ± 16.9 µg As $kg^{-1}$ grain. Comparable inorganic arsenic levels (between 100 and 400 µg $kg^{-1}$ grain) have been reported for grains without husk[55] and without bran[56] of Californian grown white rice. In fact, and in accordance with the present study, unpolished white grain has been reported to contain more arsenic compared to polished grain[57]. Nonetheless, differences in arsenic accumulation in the grain have been reported and debated when rice is grown in pots compared with fields due to maintaining a constant head of flood water in the pots rather than variable and dynamic flooding and draining conditions in the field[58–60]. Our results may thus magnify the effect of future climate change, but the forcing of increased arsenic mobility, changes in speciation and coupled physiological plant stress with increased temperature will impact rice yields and grain quality consistent with our findings[61,62].

For rice variety M206, arsenic passed the bran and entered the endosperm of the grain more extensively under future climatic conditions. In fact, higher temperatures doubled inorganic arsenic in the grain, while organic arsenic concentrations remained unchanged. This is consistent with Arao et al.[52], who showed that higher air temperature in the late ripening stage of rice increased inorganic arsenic in the grain. Overall, the change in climatic conditions did not affect the organic arsenic content in the grain, and similarly did not affect DMA in pore-water (Supplementary Fig. 7). Increased soil arsenic, however, leads to greater organic arsenic concentrations within the grain without notable change in inorganic arsenic, consistent with previous findings[58,63]. The higher soil arsenic concentration lead to a saturation of total arsenic within the grain, thus shifting the species of arsenic from organic to inorganic under elevated temperature and future climatic conditions. Governmental guidelines for arsenic levels within food, in fact, focus on restricting inorganic arsenic. Therefore, even without a change in soil arsenic concentration, climate change and most notably temperature, will cause grain inorganic arsenic levels to increase. In fact, under a future climate, more DMA is retained by the husk, preventing DMA from entering the bran. The combined impacts of high soil arsenic and high temperature, however, appear to saturate husk retention and lead to greater arsenic levels reaching the grain. Thus, overall grain toxicity to humans will increase under a shifting climate.

Collectively, our findings illustrate that for *Oryza sativa* L. variety M206, soil arsenic concentrations are major determinants of rice yield, with increasing soil arsenic levels having an overriding impact on yield decreases, while temperature combines to result in large increases in inorganic arsenic levels within rice grains. The doubling of inorganic arsenic in rice grain at any soil arsenic concentration with temperatures increases of 5 °C severely enhances the potential for human dietary exposure to arsenic, especially when considering that infant food levels for arsenic in the United States and the European Union are regulated to be <100 μg As kg$^{-1}$ grain. Extrapolating from the Californian rice variety M206 and Californian paddy soil to other varieties and soils, assuming that the direction of response to climate and soil arsenic stresses will be similar even if different in magnitude, leads to the important conclusion that rice grain yield and quality losses will result from interlinking soil arsenic and climatic drivers. Thus, factors considered for predictions of future rice productivity need to be expanded to paddy soil arsenic contents coupled with assessment of future climatic conditions. Grain quality assessments should consider not just soil arsenic but climate-associated influences on soil biogeochemical cycles that influence toxin levels in rice grains. Current endeavours of plant breeding seek either climate- or arsenic-resistant varieties, though seeking rice varieties that minimise arsenic yield impacts and managing paddy soils to limit temperature-induced changes to arsenic speciation and plant availability will be needed to limit the effects of the coupled arsenic-climate impact on rice.

## Methods

**Soil and rice characterisation**. Paddy soil and rice grains were obtained from a rice farm near Arbuckle, California (39°01′38.2″N 121°55′36.7″W, elevation: 22 m) in June 2016. Rice grains *Oryza sativa* L. sub-species japonica, cultivar Calrose, variety M206 were produced in 2015. Paddy soil texture was determined in triplicates at 20 °C by mixing ~75 g of air dry soil with 100 mL of 50 g L$^{-1}$ sodium hexametaphosphate ((NaPO$_3$)$_6$, analytical grade) and topped up to 1 L with water. The fraction of sand was quantified with a hydrometer after 40 s, silt after 2 h, and the remaining fraction was calculated as clay. Soil pH was quantified in triplicates from air-dried soil with double-distilled water at a 1:5 w/v ratio after 2, 24 and 48 h at room temperature. The elemental content of the soil was quantified with X-ray fluorescence (XRF, Spectro XePositive HE XRF Spectrometer, AMETEK, Germany) from 5 g of freeze-dried soil. The cation exchange capacity was quantified in triplicates from air-dried soils with 0.1 M BaCl$_2$ (analytical grade) at a 1:25 w/v ratio for 4 h at room temperature. Inductively coupled plasma–optical emission

spectroscopy (ICP-OES, iCAP6000, Thermo Scientific, UK) was used for quantification of the extracted elements. The total carbon and nitrogen content were quantified in triplicates from freeze-dried soils by combustion in tin foil balls (Carlo-Erba NA 1500, USA). 0.5 M HCl (analytical grade) extractable elemental fraction was quantified with ICP-OES and ferrozine for Fe(II)/Fe(III) speciation[64] after extraction of air-dried soil at a 1:40 w/v ratio at room temperature. To amend the paddy soil with arsenic, wet field soil was mixed with autoclaved double-distilled water containing 80% arsenite (NaAsO$_2$, analytical grade) and 20% arsenate (Na$_2$HAsO$_4$x7H$_2$O, analytical grade) (see Supplementary Table 1). Thus, soil arsenic concentrations varied from a background value of 7.3 mg As kg$^{-1}$ dry soil to 24.5 mg kg$^{-1}$. The soil was left to equilibrate for 4 months at the specific climatic condition. Arsenate and arsenite adsorb rapidly to soil mineral surfaces, generally reaching steady-state pore-water concentrations within hours; even with the maturing of arsenic–iron precipitates, steady-state dissolved concentrations are reached within 3 months[65,66]. Thus, our pre-incubation time allows ample time for arsenic to bind to mineral surfaces and reach conditions similar to field settings.

**Greenhouse design**. Rice was grown under controlled environmental conditions in growth chambers (polycarbonate, 1.2 × 1.8 × 1.8 m) within two larger greenhouses (Supplementary Fig. 1). Each greenhouse contained four growth chambers: one exhibiting today's climatic conditions found in Californian rice-growing regions (33 °C (average high daytime temperature during rice-growing season from May to September) and atmospheric CO$_2$ of 415 ppmv), one with future climatic conditions according to the RCP 8.5 scenario of the latest IPCC report[11] (38 °C and atmospheric CO$_2$ of 850 ppmv) and two growth chambers with either climate parameter (elevated CO$_2$ or elevated temperature). Outside air was fed into the centre of each chamber through a fan running at ~10 L of air s$^{-1}$, exchanging the entire atmosphere of the growth chamber ten times per hour. Air exited the growth chamber through a top vent. To achieve doubled atmospheric CO$_2$ contents in future climate and elevated CO$_2$ chambers, 99.9% industrial grade CO$_2$ (Praxair Inc.) was injected continuously directly into the inflowing air at the mouth of the fan. CO$_2$ flow was monitored in one chamber every second with an infra-red gas analyser (Gashound LI-820, LI-COR Inc.), and adjusted simultaneously for all four high CO$_2$ chambers using mass flow controllers (Tylan FC-260, International Power Sources Inc.) controlled by a digital to analogue converter (SDM-AO4, Campbell Scientific Inc.). The atmospheric CO$_2$ content was monitored in all chambers at two locations in between the rice plants using an infra-red gas analyser (LI-6262 CO$_2$/H$_2$O Analyser, LI-COR Inc.), which was calibrated every second week. CO$_2$ in chambers with low CO$_2$ averaged 418 ± 27 ppmv over the season and 885 ± 53 ppmv in chambers with high CO$_2$.

The temperature within each chamber was controlled by heater fans (King Electric PHM-1 1500-Watt Portable Milkhouse Heater) connected to timers (digital timer 95205, Chicago Electric); the heaters were operated within the growth chambers from the hours of 05:30 to 19:30.

Each chamber contained two water-filled basins (80 × 60 × 28 cm), in which rice pots were placed. Each water basin was heated constantly by 125 W of fully submersible aquarium heaters (JAGER TruTemp Aquarium Heater, Eheim) to 5 °C lower temperatures compared with the atmospheric daytime temperature (i.e., 27 °C and 33 °C for low and high temperature chambers, respectively). Thus, night-time atmospheric temperatures decreased to 20 °C and 22 °C for low and high temperature chambers between the hours of 19:30 and 05:30. Temperature and atmospheric CO$_2$ parameters were monitored and controlled with the programme LoggerNet 3.3.1 (Campbell Scientific Inc.), which was executed by the datalogger control unit CR10 (Campbell Scientific Inc.). An even temperature and atmospheric CO$_2$ distribution within each chamber was assured in pre-experimental climate runs of the greenhouse.

Normal daylight was supplemented with regular lights in the greenhouse between the hours of 06:00 and 19:00 to support naturally occurring sunlight and to minimise shading within each chamber. The photosynthetic photon flux density was between 300 and 600 μmol m$^{-2}$s$^{-1}$ (equivalent to a photosynthetic active radiation of 65–130 W m$^{-2}$) throughout the day, depending on the prevailing weather during the season.

**Experimental design**. Each growth chamber hosted microcosms with natural and amended soil arsenic levels, which were placed into either of the two water-filled basins (see Supplementary Fig. 1). The water-filled basins served two purposes; one to provide a lower and more constant temperature to the soil compared with the atmospheric temperature, and second to maintain temperature differences in low and high atmospheric temperatures overnight. Rice pots (high-density poly-ethylene, 14-cm diameter, 18-cm height) were filled with ~3.4 kg of water-logged paddy soil to 4 cm below the rim. Rhizon samplers (19.21.25, Rhizosphere Research Products) were placed horizontally midway in each pot. Soil was flooded with tap water, pH 7, and irrigated continuously with pumps through a drip irrigation system using water from the basins.

Rice variety M206 was germinated in sterilised tap water under the different climatic conditions. After 1 week of germination, seedlings of similar shoot and root lengths were placed into pots. The soil was fertilised with urea (CON$_2$H$_4$, analytical grade) using 185 kg nitrogen ha$^{-1}$ at the beginning of the season, after 4.5 weeks of growth and at panicle initiation, according to practices performed by Californian farmers. Three plants were planted per pot and eight pots per

environmental condition, two independently run greenhouses with four climate chambers each. To minimise light, temperature and atmospheric $CO_2$ biases due to placement of plants within and across chambers, pots were moved around within basins once a week, and chambers were switched every 3–4 weeks.

**Pore-water analysis**. Pore-water geochemistry was examined regularly by drawing pore-water through rhizon samplers into acid-washed, butyl-stoppered and evacuated glass vials. The pore-water was filtered through 0.2 -μm syringe filters in an anoxic chamber (96% $N_2$/4% $H_2$ atmosphere) and diluted in 2% nitric acid (analytical grade) for total elemental (As, Fe) quantification with ICP-OES. Pore-water organic and inorganic carbon was quantified using the total organic carbon method (TOC) on a Shimadzu TOC-L analyser with in-line acidification (phosphoric acid). Once a month, an aliquot of the unfiltered pore-water was used to determine the pore-water pH, which remained constant during growth (data not shown). Arsenic speciation in the pore-water was analysed using an arsenic speciation cartridge (anion exchanger, Metalsoft, USA) and double-checked by ion chromatography (Dionex ICS-3000, Dionex Corp., USA, PRPX-100 column using a 10–40 mM $NH_4H_2PO_4$ pH 5.6 gradient) coupled to ICP-MS (XSeries2, Thermo Fisher Scientific Inc., USA) (IC–ICP-MS). Using IC–ICP-MS, we found that DMA was present in pore-water, but only contributed to <3% to total arsenic, which was negligible (Supplementary Fig. 7). Thus, we used the simpler arsenic speciation cartridge method, discriminating more easily between charged (arsenate) and non-charged (arsenite) arsenic species. Thereby, DMA and MMA are most likely summarised with arsenate. To do so, 2–3 mL of pore-water were diluted in 13–14 mL of anoxic double-distilled water and run through a water-flushed arsenic speciation cartridge, which retains the charged arsenic (likely mainly arsenate). Uncharged arsenic (arsenite) in the filtered pore-water was quantified with ICP-OES. The amount of arsenate was calculated by subtracting the measured concentration of arsenite from the total pore-water arsenic.

**Plant analysis**. Plant growth over time was monitored weekly by quantifying plant height, tiller number and panicle number. At harvest, each rice plant was cut-off 2 cm above the water level, packaged into butcher paper and dried at 40 °C in the dark for 4 weeks. Dried plant material was separated into filled spikelets, empty spikelets, panicles and straw (combined leaves and stems), and each fraction was weighed and counted. The yield is defined as the weight of filled spikelets. The harvest index is defined as the grams of filled spikelets produced per plant compared with the grams of vegetative, aboveground tissue produced per plant. Individual grain weight, percentage of filled grains and the number of spikelets per panicle were calculated from weight and count data. Husks were removed from grains by rubbing filled spikelets between two plastic crusher plates. Grains, empty spikelets, husks and green biomass were ground (conventional mechanical grinding followed by liquid nitrogen grinding when needed). Ground grains were stored in the freezer at −80 °C to minimise arsenic speciation changes. Between 0.05 and 0.5 g of different plant tissue was extracted with 3 mL of 65% nitric acid ($HNO_3$, analytical grade) and 2 mL of 30% $H_2O_2$ (analytical grade) in a microwave digester (CEM Mars 6 Digester) with a 15-min ramp phase to 15 min at 180 °C, followed by a 30 min cool-down phase. Extractants were diluted in double-deionised water, and elemental contents were quantified with inductively coupled plasma mass spectrometry (ICP-MS) (XSeries2, Thermo Fisher Scientific Inc., USA)[67,68]. For arsenic speciation, 1 g of ground rice grain was extracted with 10 mL of 0.28 M $HNO_3$ for 90 min at 95 °C in a heating block (Digi PREP Jr, SCP Science, Canada) followed by a 20 min cool-down phase. Extractants were filtered through a 0.45 -μm filter (SCP Science), and arsenic species were separated and analysed by IC–ICP-MS using the method described above for pore-water speciation. The European Reference material ERM-BC211 was used for rice speciation and TMDA-54.4 (Environment Canada) for calibration accuracy. Every 15th sample was spiked with 2 ppb and 5 ppb As(V), As(III), DMA and MMA to check for the recovery of each of these species. The first sample that was analysed was re-analysed at the end of the run to check for species transformation before analysis and signal drift of the instrument. Further, a 10 ppb calibration standard was measured after every 15th sample to check for instrument signal drift. Mean and standard deviations were calculated of biological replicates and unpaired, two-tailed Welch t tests were performed across all soil arsenic and climatic conditions.

**Microbial analysis**. Rhizosphere soil aliquots were taken sterilely from soil around plant roots midway at the same height as rhizon samplers were placed within the pots, shock-frozen in liquid nitrogen, and stored at −80 °C. The total DNA was extracted from 0.25 g of two frozen samples and using the PowerSoil® DNA Isolation Kit (Qiagen, Germantown, MD, USA). The quality and quantity of DNA were verified on 1% (w/v) ethidium-bromide-died agarose gels and fluorometric quantification with Qubit® 2.0. The 16S rRNA gene copy numbers of the total Bacteria were amplified and quantified with qPCR using the SsoAdvanced Universal SYBR Green Supermix (Bio-Rad Laboratories, Hercules, CA, USA) on a StepOne™ Plus cycler (Applied Biosystems, Waltham, MA, USA). As a standard, a 16S rRNA gene fragment from *Gemmatimonas* sp. was amplified from the rhizosphere DNA extracts with general bacterial primers GM3 (8F, (5′-AGAGTTT GATCMTGGC-3′)[69] and 1392-R (5′-GACGGGCGGTGTGTRCA-3′)[70] and cloned into the plasmid vector pGEM®-T easy (Promega Corporation, Madison, WI, USA). In a 10 μL of reaction volume, 1 μL of 100-fold diluted DNA extract or a tenfold dilution series of the standard plasmid DNA were used with 1× SsoAdvanced Universal SYBR Green Supermix, 75 nM of primer 341-F (5′-CCTACGG GAGGCAGCAG-3′)[71] and 225 nM of primer 797-R (5′- GGACTACCAGGGTA TCTAATCCTGTT-3′)[72]. The qPCR programme ran with 3 min at 98 °C, 40 cycles of 15 s at 98 °C and 30 s at 60 °C, and followed by melting curve analysis. The data analysis was performed using the StepOne™ 2.3 software. Each of three independent DNA extractions were measured in triplicates.

**X-ray fluorescence imaging**. The distribution of arsenic and other elements across husked rice grains was visualised with X-ray fluorescence (XRF) mapping using beamline 2–3 at Stanford Synchrotron Radiation Lightsource (SSRL). The beamline receives X-rays from a 1.3 Tesla Bend Magnet, and it is equipped with a double-crystal Si (111) monochromator for energy selection, a vortex silicon drift detector and ionisation chambers. The sample is positioned in a 45° angle to the incoming beam and another 45° angle to the detector (90° angle between incoming beam and detector). The Kirkpatrick-Baez mirror system achieves a beam size of ~2 × 2 microns. The beam was calibrated to the arsenic K-edge of an arsenate standard, and maps were run at the energy for arsenite, dimethylarsinic acid (DMA) and total arsenic, rastering at a 0.013-mm step size and a 45-ms dwell time. Mapping energies for arsenite and DMA were chosen as they contribute >95% to the respective inorganic and organic arsenic fractions in the grain (previously verified with IC–ICP–MS in grain extracts). Rice grains were split in half on the longitudinal axis of the grain and fixed on glass slides. One single replicate grain was run per climatic and soil arsenic condition. Using the software SMAK[73], arsenic fluorescent counts were quantified to an arsenic standard of known concentration (47 μg As $cm^{−2}$) and expressed in mg As $kg^{−1}$ grain for a sample thickness of 1 mm and a grain density of 0.8 kg of grain $L^{−1}$. Average fluorescent counts for DMA and arsenite were obtained for husks, brans and endosperm. DMA and arsenite ratios from the husk to bran, bran to endosperm and within husk, bran and endosperm were calculated.

**Data analysis**. Mean and standard errors were calculated of all data set. The data were verified for normal distribution with the Kolmogorov–Smirnov Test. Standard Student's t tests were combined with a two-factorial ANOVA analysis to display significant differences in means of data and illuminate significant interactions between climatic condition and soil arsenic (Supplementary Table S2).

## Data availability

The data sets generated during and/or analysed during the current study are also available from the corresponding authors on reasonable request.

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

## Acknowledgements

We thank Todd Tobeck, Lance Cabalona, Ismael Villa, Theo Van de Sande for greenhouse support. We also thank Maegen Simmonds, Guangchao Li, Doug Turner, Mariejo Plaganas, Sindhu Goli, Howard Tang and Fendorf lab members for help in the laboratory and greenhouse. Greenhouse space was graciously provided by Christopher Field and Joe Berry at Carnegie Institution for Science, Department of Global Ecology. This work was financed through a Marie Skłodowska Curie Action fellowship by the Horizon 2020 of the European Commission to EMM (RACe). We thank Sam Webb, Sharon Bone and Christian Dewey for X-ray fluorescence elemental mapping. Use of the Stanford Synchrotron Radiation Lightsource, SLAC National Accelerator Laboratory, is supported by the U.S. Department of Energy, Office of Science, Office of Basic Energy Sciences under Contract No. DE-AC02-76SF00515. The SSRL Structural Molecular Biology Program is supported by the DOE Office of Biological and Environmental Research, and by the National Institutes of Health, National Institute of General Medical Sciences (including P41GM103393). The contents of this publication are solely the responsibility of the authors and do not necessarily represent the official views of NIGMS or NIH.

## Author contributions

This work was conceptualised by E.M.M. and S.F. Greenhouse work and plant and soil geochemical measurements were planned by E.M.M. with input from S.F. and carried out by E.M.M. and T.W. Synchrotron work was carried out by E.M.M. Rice grain arsenic speciation measurements were carried out by B.P.F. and C.K. The paper and supporting information were written by E.M.M. with primary input from S.F. and additional input from all co-authors.

## Competing interests

The authors declare no competing interests.
