## [Peer Review File · Nature Communications]

Reviewers' comments:

Reviewer #1 (Remarks to the Author):

The authors assessed the effects of future climate (increased temperature and/or elevated CO₂) on soil arsenic behaviour and plant response including grain yield and As accumulation using Californian *Oryza sativa* L. variety M206, grown on Californian paddy soil. Though this work is interesting, it needs major revision. I have few issues with this MS. for example.

I think the As uptake by plants not only depends on the quantity of As in soil but also the other factors including microbial diversity and activity, plant-microbe interactions, plant physiological changes due to climatic factors including temperature and CO₂ levels and physicochemical features of rhizosphere soils, etc. Authors have discussed with published literatures about the effects of increased temperature on microbial activity, As and Fe speciation, however, the supporting data regarding the interactive effects of climatic factors (increased temperature and/or elevated CO₂) and plants on microbial diversity/activity in As contaminated soils is necessary in order to explore how future climate influence As uptake in plants. Moreover, several recent studies also described increased C flow as a result of plant root exudation under elevated CO₂ condition lead to an increase or decrease in microbial diversity and activity. Thus I strongly recommend the authors to include the data about the effects of increased temperature and/or elevated CO₂ on microbial diversity/activity and the characteristics features of rhizosphere soil.

Similarly the data (Fig 2a) indicates that the future climate increase As accumulation in the grains grown in low As soils, but have no significant effect on the accumulation of As in the grains grown in high As soils. So the authors should provide proper explanation on how future climate alter the accumulation of As in the grains.

Similarly authors have provided the data (Fig 3C) that elevated CO₂ showed a decrease in As III levels in As low and high soil (88 and 64 % respectively), as compared with today's climate (92 and 75 % respectively), but no proper explanation on how CO₂ influence As V and As III levels in pore water.

Reviewer #2 (Remarks to the Author):

Muehe et al. used a glasshouse pot experiment to investigate the effect of global climatic changes (warming and elevated CO₂) combined with elevated soil arsenic on rice yield and As content. The striking result from the study is that future climate (+5 °C & 800 ppmv [CO₂]) can cause a large loss of grain yield and increase grain As concentration. This result provides an important warning for the potential threat from global climate changes.

General comments:

1. According to the IPCC Climate change report 2014, it is very likely that actual atmospheric [CO₂] will reach 570 ppmv (compared to the present ambient [CO₂] around 370-400 ppmv) and temperature increase around 1.5-2 °C by the end of this century. The climate conditions used in Muehe et al are far from this projection for 2100. The authors should discuss whether a more realistic future climate change would result in the large effects observed in their study.
2. As noted by the authors, grain As concentrations from their glasshouse pot experiment are comparable to the reported As concentrations from other pot experiments, but appear to be much higher than the reported values (0.1-0.5 mg/kg) from As-contaminated paddy fields. A likely reason is that the soil was flooded continuously in the pot experiment, thus increasing the As availability, whereas paddy fields often undergo periodic flooding and draining. The experimental conditions used in the pot study might have magnified the effect of future climate change, a possibility that should be discussed. In fact, there are FACE experiments with paddy rice around the world. It would strengthen this manuscript considerably if rice samples from some of the FACE experiments could be obtained and analysed.
3. Given that organic As (DMA) accounts for a large proportion of the total grain As (Figure 2A), it would be of value to see the temporal changes in the porewater DMA concentration and how it is affected by temperature and eCO₂.
4. Statistics: it would be useful to see the ANOVA results, especially with regard to the significance of the interactions between As and climate conditions.
5. I find the descriptions of the results, listing individual values, rather tedious and repeating what are already presented in the Figures/Tables.

Response to Reviewers' Comments:

Reviewer comments are provided in black font and our response to each comment and changes made within the revised manuscript are indicating by a blue font color. We also added five citations with the requested changes.

Reviewer #1:

The authors assessed the effects of future climate (increased temperature and/or elevated CO₂) on soil arsenic behaviour and plant response including grain yield and As accumulation using Californian *Oryza sativa* L. variety M206, grown on Californian paddy soil. Though this work is interesting, it needs major revision. I have few issues with this MS. for example.

We thank the reviewer for his/her helpful comments and believe to have addressed all concerns.

I think the As uptake by plants not only depends on the quantity of As in soil but also the other factors including microbial diversity and activity, plant-microbe interactions, plant physiological changes due to climatic factors including temperature and CO₂ levels and physicochemical features of rhizosphere soils, etc. Authors have discussed with published literatures about the effects of increased temperature on microbial activity, As and Fe speciation, however, the supporting data regarding the interactive effects of climatic factors (increased temperature and/or elevated CO₂) and plants on microbial diversity/activity in As contaminated soils is necessary in order to explore how future climate influence As uptake in plants. Moreover, several recent studies also described increased C flow as a result of plant root exudation under elevated CO₂ condition lead to an increase or decrease in microbial diversity and activity. Thus I strongly recommend the authors to include the data about the effects of increased temperature and/or elevated CO₂ on microbial diversity/activity and the characteristics features of rhizosphere soil.

We fully agree with the reviewer and recognize that plant-microbe interaction and rhizosphere microbial activity is crucial for the outcome of arsenic and climate impacts on rice performance. We have therefore included microbial analysis that includes qPCR data on bacterial 16S rRNA gene copies and measures of carbon allotment and utilization to the supplemental file (Figure S6) and amended the complementary methods (lines 494-513), results (lines 244-263) and discussion (lines 279, 295-308) to the manuscript.

Similarly the data (Fig 2a) indicates that the future climate increase As accumulation in the grains grown in low As soils, but have no significant effect on the accumulation of As in the grains grown in high As soils. So the authors should provide proper explanation on how future climate alter the accumulation of As in the grains.

We have provided an explanation on why total arsenic does not increase in the grain but that the speciation changes with climatic change within the revised manuscript (lines 329-332).

Similarly authors have provided the data (Fig 3C) that elevated CO₂ showed a decrease in As III levels in

As low and high soil (88 and 64 % respectively), as compared with today's climate (92 and 75 % respectively), but no proper explanation on how CO₂ influence As V and As III levels in pore water.

Thank you for this comment, we have clarified the statement and separated temperature and atmospheric CO₂ data in line 301-308.

Reviewer #2 (Remarks to the Author):

Muehe et al. used a glasshouse pot experiment to investigate the effect of global climatic changes (warming and elevated CO₂) combined with elevated soil arsenic on rice yield and As content. The striking result from the study is that future climate (+5 °C & 800 ppmv [CO₂]) can cause a large loss of grain yield and increase grain As concentration. This result provides an important warning for the potential threat from global climate changes.

We thank the reviewer for providing very insightful feedback to this manuscript and have sought to address each point.

General comments:

1. According to the IPCC Climate change report 2014, it is very likely that actual atmospheric [CO₂] will reach 570 ppmv (compared to the present ambient [CO₂] around 370-400 ppmv) and temperature increase around 1.5-2 °C by the end of this century. The climate conditions used in Muehe et al are far from this projection for 2100. The authors should discuss whether a more realistic future climate change would result in the large effects observed in their study.

As of 2017, IPCC has declared the worst-case climate projection of 2014 as the most likely and possibly even a moderate scenario to occur given current global trends. Thus, the most likely scenario project by the IPCC is indeed an increase in [CO₂] to 900 ppmv and nearly 5 °C increase in average temperature. We have elaborated and clarified this statement in lines 86-94.

2. As noted by the authors, grain As concentrations from their glasshouse pot experiment are comparable to the reported As concentrations from other pot experiments, but appear to be much higher than the reported values (0.1-0.5 mg/kg) from As-contaminated paddy fields. A likely reason is that the soil was flooded continuously in the pot experiment, thus increasing the As availability, whereas paddy fields often undergo periodic flooding and draining. The experimental conditions used in the pot study might have magnified the effect of future climate change, a possibility that should be discussed. In fact, there are FACE experiments with paddy rice around the world. It would strengthen this manuscript considerably if rice samples from some of the FACE experiments could be obtained and analysed.

We thank the reviewer for this thoughtful comment. As we discuss within the manuscript (lines 316ff), each of the approaches for simulating future climates have attributes and limitations. A comprehensive analysis across all types of physical simulations, from greenhouse to FACE experiments, would be a useful endeavor. We also strongly believe a future study should take on this task. At present, and as noted by the Reviewer, this study "provides an important warning for the potential threat from global climate changes". Conducting a robust comparison study, with statistical validity, would likely take a couple years and thus delay the dissemination of these important findings. We therefore believe moving our manuscript forward is important while initiating a secondary study that seeks to integrated

the various approaches for climate simulation.

3. Given that organic As (DMA) accounts for a large proportion of the total grain As (Figure 2A), it would be of value to see the temporal changes in the porewater DMA concentration and how it is affected by temperature and eCO₂.

We did quantify DMA and MMA in the pore-water at harvest; they account for less than 3% of DMA in the pore-water, and we observe no prominent effect of either temperature or atmospheric CO₂. We now provide the data in Figure S7. As these amounts were negligible, we only followed inorganic arsenic speciation with the anion exchanging arsenic speciation cartridges as already presented in Figure 2. This information is also provided in the method section on lines 451-462. We also discussed the apparent difference between high DMA in grain compared to low DMA in pore water in lines 322-388 and added 3 citations to support the discussion.

4. Statistics: it would be useful to see the ANOVA results, especially with regard to the significance of the interactions between As and climate conditions.

We now included a two factorial ANOVA analysis in Supplementary table 2 for yield, grain As levels, husks As levels and As retention by husk. Details are further added to the method (lines 533-537), results sections (lines 125, 188).

5. I find the descriptions of the results, listing individual values, rather tedious and repeating what are already presented in the Figures/Tables.

We cut down the results section by taking out lists of individual values of data shown in the supplement and restructured the entire results section so that the same order of environmental condition is described to cause less confusion.

REVIEWERS' COMMENTS:

Reviewer #1 (Remarks to the Author):

The MS (NCOMMS-19-15887B) has been substantially improved by the authors as per the reviewer's comments, thus I would recommend the article for publication in its present form.

Reviewer #2 (Remarks to the Author):

The authors have addressed my comments adequately. I have two minor comments on this revised version:

- 1). Please indicate in the Abstract that the study was based on a greenhouse pot experiment.
- 2)L127, a p value of 0.07 means no significant (or weak) interactions between soil arsenic and climatic conditions.

Response to Reviewers Comments

Rice production threatened by coupled stresses of climate and soil arsenic
(NCOMMS-19-15887B)

Reviewer 1

No further changes were suggested

Reviewer 2

Comment 1. *Please indicate in the Abstract that the study was based on a greenhouse pot experiment.*

Response and Changes Made: We now indicate in the abstract that our findings are based a greenhouse experiment.

Comment 2. *L127, a p value of 0.07 means no significant (or weak) interactions between soil arsenic and climatic conditions*

Response and Changes Made: We agree that a p value of 0.07 indicates a weaker interaction. We have changed the text to represent the value more clearly.